# The Effect of PCL Addition on 3D-Printable PLA/HA Composite Filaments for the Treatment of Bone Defects

**DOI:** 10.3390/polym14163305

**Published:** 2022-08-13

**Authors:** Elin Åkerlund, Anna Diez-Escudero, Ana Grzeszczak, Cecilia Persson

**Affiliations:** 1Division of Biomedical Engineering, Department of Materials Science and Engineering, The Ångström Laboratory, Uppsala University, 752 37 Uppsala, Sweden; 2Ortholab, Department of Surgical Sciences, Rudbeck Laboratory, Uppsala University, 752 37 Uppsala, Sweden

**Keywords:** biodegradable, polylactic acid (PLA), polycaprolactone (PCL), hydroxyapatite, 3D printing, fused filament fabrication (FFF), fused deposition modelling (FDM), mechanical properties, accelerated degradation study

## Abstract

The still-growing field of additive manufacturing (AM), which includes 3D printing, has enabled manufacturing of patient-specific medical devices with high geometrical accuracy in a relatively quick manner. However, the development of materials with specific properties is still ongoing, including those for enhanced bone-repair applications. Such applications seek materials with tailored mechanical properties close to bone tissue and, importantly, that can serve as temporary supports, allowing for new bone ingrowth while the material is resorbed. Thus, controlling the resorption rate of materials for bone applications can support bone healing by balancing new tissue formation and implant resorption. In this regard, this work aimed to study the combination of polylactic acid (PLA), polycaprolactone (PCL) and hydroxyapatite (HA) to develop customized biocompatible and bioresorbable polymer-based composite filaments, through extrusion, for fused filament fabrication (FFF) printing. PLA and PCL were used as supporting polymer matrices while HA was added to enhance the biological activity. The materials were characterized in terms of mechanical properties, thermal stability, chemical composition and morphology. An accelerated degradation study was executed to investigate the impact of degradation on the above-mentioned properties. The results showed that the materials’ chemical compositions were not affected by the extrusion nor the printing process. All materials exhibited higher mechanical properties than human trabecular bone, even after degradation with a mass loss of around 30% for the polymer blends and 60% for the composites. It was also apparent that the mineral accelerated the polymer degradation significantly, which can be advantageous for a faster healing time, where support is required only for a shorter time period.

## 1. Introduction

Additive manufacturing (AM) has been applied in the automobile and aeronautical industries since the beginning of the 1980s. However, its application in medical industries is more recent and yet to be fully explored. In the past few decades, the combination of high-resolution imaging with AM has enabled the development of customized implants and surgical devices [1]. However, much work remains to be undertaken in the development of new biomaterials that are possible to manufacture using AM.

Due to the quick processing time as well as its low cost, fused filament fabrication (FFF), or fused deposition modelling (FDM), stands as an attractive 3D-printing technique for fabricating medical devices [1]. FFF manufactures 3D objects by melting thermoplastic polymer filaments while extruding them through a heated nozzle and depositing the melted materials onto a build plate with high geometrical accuracy [1,2,3]. Recently, efforts have been put into investigating and developing new synthetic materials with good printability, aiming at their application in bone tissue engineering. In this regard, biocompatible materials [2] have attracted much attention; such is the case for thermoplastic polyesters. Polymer-based scaffolds aimed at regenerative bone tissue engineering are envisioned as temporary scaffolds with the potential to assist bone regeneration. Therefore, bioresorbable materials that support new bone tissue formation while they are actively or passively resorbed without eliciting detrimental effects to bone healing [4], such as an excessive inflammatory response, are highly sought.

Polylactic acid (PLA) is one thermoplastic polyester applied in clinics as resorbable support materials for bone and ligament fixation [5,6], with high interest for FFF due to its good printability, biocompatibility and bioresorbability [7]. It also possesses relatively high mechanical properties and can be fabricated from renewable resources, such as corn [8,9,10,11]. Despite the biocompatibility and degradability of PLA, its bioactivity is scarce and the development of composite materials, including mineral phases, that further resemble tissue composition is of great interest. For instance, hydroxyapatite (HA), the mineral phase of bone tissue, has been found to increase the bioactivity of the material [12,13,14]. HA itself possesses bioactivity and biocompatibility; it can stimulate bone growth and promote adhesion of tissues [7,15]. However, both of these materials are brittle [9,12], which limits their applications in load-bearing bone approaches. To increase the toughness, it is possible to add plasticizers, rigid fillers and copolymers to obtain a polymer composite that can improve the mechanical properties [8,9,10]. For instance, polycaprolactone (PCL), which is another FDA-approved [7,16] biocompatible and bioresorbable polyester, with promising properties for medical devices, has been reported to improve the toughness of brittle PLA, which may result in a polymeric matrix more suitable for bone applications [9,10,17,18].

Additionally, degradation rates and by-products of biomaterials are paramount in regenerative tissue applications. Degradation can occur actively by cellular activity and the physiological environment at the implantation site [19] or passively, driven by the physico-chemical properties in the biomaterial itself [20,21]. The degradation rates of such polymers depend on several factors, the main ones being the molecular weight, the degree of crystallinity and porosity or surface area [22]. Environmental factors, such as the surrounding temperature, pH and mechanical stimuli, can also play a role in scaffold degradation. For instance, PLA has shown degradation rates in vivo ranging from 12 months to 5 years, depending on its crystallinity [23]. Generally, PCL exhibits lower degradation rates than PLA, accounting for up to four years in some conditions [2,17]. It is, however, possible to increase the degradation rates of polymers by the addition of mineral phases such as HA [17].

Recently, the development of customized composite and blend polymers has been explored [24], including their application in the FFF technique [25,26,27,28,29]. Most of the studies focused on combining polymer matrices with bioactive minerals to improve their bioactivity, degradation and cell compatibility. Previous works, for instance, focused on combining PLA with HA or PCL with HA or similar calcium-phosphate-based ceramics. Although the use of PCL has been widely explored using other AM techniques, there is a lack of studies exploring the combination of PCL and PLA with HA using the FFF technique and, in particular, its degradation behavior. This novel approach to obtain customizable filaments to be applied in cost-effective FFF can serve as an alternative to improve not only the development of patient specific geometries but also to tune their physico-chemical performance to be adjusted to the implant-site requirements. For instance, customizable mechanical properties, depending on the blending composition or the composite formulation, can be adjusted, in addition to tunable degradation rates for different bone location requirements.

In this study, we aimed at exploring the combination of these two biocompatible polymer matrices with bioactive mineral, such as HA, with the aim of providing customized materials for FFF of potential bone implants. The combination of these materials was investigated with the aim to tune the properties to produce materials suitable for bone-repair applications, with a special focus on their degradation behavior, including mechanical and chemical stability.

## 2. Materials and Methods

### 2.1. Materials

Two biodegradable polymers were used, polylactic acid (PLA; transparent filament; 2.85 mm; 3D4Makers, Haarlem, Noord-Holland, The Netherlands) and polycaprolactone (PCL; Capa™ 6800; pellets; MW = 80000 g/mol; The Perstorp Group, Malmö, Sweden) (PCL; Facilan™ PCL100; filament; 2.85 mm; ElogioAM 3D materials, Haarlem, Noord-Holland, The Netherlands). Hydroxyapatite powder (HA; MW = 310.18 g/mol; Merck, Rahway, New Jersey, USA) was used as mineral filler and sieved below 75 µm before use [14]. Dichloromethane (≥99%; laboratory reagent grade; MW = 84.93 g/mol; Fisher Scientific, Waltham, Massachusetts, USA) was used as solvent to prepare the composite filaments. Different blend compositions were studied combining PLA and PCL, with and without mineral HA incorporation, as depicted in Table 1. For the composites with HA incorporation, 15 wt% of HA was added but the ratio between PLA and PCL content was kept at 90:10, 80:20 and 70:30. The samples’ designations were chosen to represent this ratio between PLA and PCL and to facilitate the comparison with the blends without HA incorporation, rather than repeating their respective compositions in wt%. Pure PLA and PCL were used as control samples and printed directly from the commercial filaments.

### 2.2. Blends and Composite Blends Preparation

PLA-PCL blends were prepared by cutting the PLA filament into small pieces and dissolved into dichloromethane for approximately 1 h. Afterwards, the corresponding amount of PCL pellets was added and further left under agitation until complete homogenization (4 h). The polymer blend was then cast into a crystallizer and left for solvent evaporation at room temperature for at least 20 h. Afterwards, the blend films were collected, cut and stored in desiccator until their extrusion. The composite counterparts were prepared similarly as aforementioned. The mineral HA was dispersed in the dichloromethane for 15 min prior to the addition of the polymers.

All films were finely shredded (SHR3D IT; 3devo; Utrecht, The Netherlands) prior to extrusion. The flakes were fed into a single-screw extruder (PRECISION 350; 3devo; Utrecht, The Netherlands) to obtain 2.85 mm thick filaments. The setting parameters for extrusion varied depending on the material composition (Appendix A).

### 2.3. 3D Printing

An FFF printer (Ultimaker S5; Ultimaker, Utrecht, Netherlands) was used to print the filaments. Two different print cores were used, 0.4 mm brass (Ultimaker Print Core AA; Ultimaker, Utrecht, The Netherlands) and 0.6 mm ruby (Ultimaker Print Core CC Red 0.60; Ultimaker, Utrecht, The Netherlands), for the blends and the composites blends, respectively. Dense, full infill density (100%), cylinders of 6 mm in diameter and 12 mm in height were printed with a layer height of 0.1 mm. Further, 100% fan speed was used for all materials. Pure PCL also required an external fan, placed in front of the build plate, and 3D lac (3DLAC Spray Glue; Zamora, Spain) was used as adhesion improver on the build plate for all materials except pure PCL which required Kapton tape. The printing parameters are summarized in Table 2.

### 2.4. Chemical and Morphological Evaluation

Chemical characterization was performed by Attenuated Total Reflectance-Fourier Transform Infrared Spectroscopy (ATR-FTIR; Tensor 27; Bruker, Billerica, MA, USA). A diamond crystal was used and each spectrum was recorded with a total of 64 scans, a resolution of 4 cm^−1^ and between a range of 400 and 4000 cm^−1^. Phase composition was assessed by X-ray diffraction (XRD) using a D8 Twin-Twin (D8 Advance; Bruker, MA, USA) diffractometer with a copper anode, operating at 40.0 kV and 40.0 mA, from 5 to 70° (2θ), using a step size of 0.018° and 0.50 sec per step. The patterns obtained for HA powders were compared to hydroxyapatite (JCPDS 01-074-0565) and monetite (JCPDS 04-009-3755).

Morphological characterization of the cast films and extruded filaments was performed using scanning electron microscopy (SEM; Hitachi tabletop microscope TM1000; Hitachi, Chiyoda, Tokyo, Japan) with an acceleration voltage of 15 kV. Cross sections of fracture surfaces of the films and the filaments, as well as the film surfaces, were studied on both the polymer blends and the composites. All samples were sputtered with gold/palladium prior to analysis by using a Thermo VG Scientific POLARON SC7640 Sputter Coater (Quorum Technologies, Lewes, UK).

### 2.5. Thermal Characterization

Thermal characterization was carried out by a differential scanning calorimeter (DSC; TA instruments Q2000; TA instruments, New Castle, New Castle County, Delaware, USA) and by thermogravimetric analysis (TGA; TA instruments Q500; TA instruments, New Castle, New Castle County, Delaware, USA). DSC was performed using a heat–cool–heat procedure, with a heating rate of 10 °C/min until 200 °C and a cooling rate of 5 °C/min until −100 °C, under a nitrogen flow of 50 mL/min. All samples were analysed in Aluminium hermetic pans and sample weight varied between 6 and 10 mg. The second heating scan was used to evaluate the thermal properties for each sample. TGA measurements were performed in Platinum pans at a heat rate of 10 °C/min, from room temperature to 800 °C, under a nitrogen flow of 50 mL/min. TGA sample weight varied between 17 and 23 mg.

### 2.6. Mechanical Characterization

Mechanical characterization was performed on the 3D-printed cylinders by a uniaxial compression test using a Shimadzu AGS-X universal testing machine (Shimadzu, Kyoto, Japan). Compliance correction was used by measuring the stiffness of the set-up using a 10kN cell, data which were later used in the calculations. Prior to testing, all specimens were pre-loaded with a force corresponding to 2–4% strain [30]. Samples were tested at a cross-head speed of 1 mm/min. Further, 6–11 specimens for each type of material were used and the elastic modulus and compressive strength were reported. The elastic modulus was determined by taking the slope of the initial linear region of the stress–strain curve, with any toe region neglected, while the compressive strength was taken as the yield point (if any) or as the 2% offset load (whichever occurred first) in accordance with ISO 5833:2002.

### 2.7. Degradation Studies

An accelerated degradation study was performed on the printed compositions as previously described [2]. Briefly, 0.1M NaOH solution (Sigma-Aldrich, Saint-Louis, MO, USA) was used as degradation media. Cylinders were individually immersed in 10 mL of solution and kept at physiological temperature (37 °C) over a four-week period. The media were refreshed every third day. The cylinders were removed at 6, 24, 48 and 72 h and at 7, 14, 21 and 28 days. After removal from the degradation media, the cylinders were thoroughly rinsed in MilliQ water and dried overnight at 37 °C. The percentage of weight loss (*W_L_*) of all specimens was determined by using the following equation:(1)$$W_{L}\left(\%\right)=\frac{W_{0}-W_{F}}{W_{0}}\times 100,$$
where *W*_0_ represents the initial mass of the cylinders and *W_F_*, the final mass of the cylinders after immersion in NaOH solution for each specific time point. Triplicates were used for hour interval measurements (6, 24, 48 and 72 h) and eight replicas were used for the daytime intervals (1, 14, 21 and 28 days).

Compositional changes in the cylinders were analysed by FTIR, XRD, DSC and TGA. Morphological changes on the surfaces were observed by SEM (Zeiss Merlin Gemini Scanning Electron Microscope; Zeiss, Oberkochen, Germany). The mechanical behaviour of degraded samples was investigated at 7-, 14-, 21- and 28-day time points using six specimens per time point. A 5 kN load cell was used, without pre-loading, with a cross-head speed of 1 mm/min. The elastic modulus and yield point were determined from the obtained stress–strain curves as described in Section 2.6.

### 2.8. Statistical Analysis

Statistical analysis was conducted using SPSS Statistics (IBM, Armonk, NY, USA). The test results are presented as mean and standard deviation and were considered significant if *p* < 0.05. First, Levene’s test was used to assess homogeneity of variances. If Levene’s test was non-significant, an analysis of variance (ANOVA) was performed to analyse the differences among the group means. If the ANOVA was significant, Scheffe’s post hoc test was conducted. If Levene’s test was found significant, Welch’s test was conducted to analyse the differences among group means. Then, Tamhane’s post hoc test was carried out to analyse the differences between pairs of groups.

## 3. Results

### 3.1. Chemical and Morphological Evaluation of Blends and Composite Blends

Spectroscopic analyses of the filaments for each composition are illustrated in Figure 1. CH_2_ stretching bands were shown at 2995 cm^−1^ and 2950 cm^−1^ for pure PLA and at 2947 cm^−1^ and 2865 cm^−1^ for pure PCL, whereas a C=O stretching band appeared at 1750 cm^−1^ and 1720 cm^−1^ for PLA and PCL, respectively. Moreover, the CH_3_ bending band was found at 1453 cm^−1^ for PLA and C–O–C stretching bands at 1180 cm^−1^ and 1082 cm^−1^ for PLA and 1171 cm^−1^ for PCL [4,31,32,33]. These bands were also found in the blends. The HA mineral incorporation was evidenced in the composite blends by the bands observed at 1026 cm^−1^, 601 cm^−1^ and 561 cm^−1^, corresponding to vibrations in the phosphate group (PO_4_^3-^) in HA [34,35].

In addition to the chemical changes observed by FTIR, the morphology of the materials assessed by SEM varied (Figure 2). Cross-sectional analysis of the filaments indicated a smooth-layer-structured morphology for PLA, a smooth fibrous structure for PCL, a porous morphology for the blends and a rougher morphology for the composite blends, due to the addition of mineral into the polymer matrix. Furthermore, increasing amounts of PCL in the blends gave the cross section a more ductile fibrous appearance as well as larger pores [35].

### 3.2. Thermal Characterization

DSC measurements were performed to investigate the thermal behaviour of PLA and PCL in the processed blends and composite blends. Table 3 summarizes the main thermal characteristics for all materials. All blends and composite blends presented a melting peak for PCL around 55 °C while the glass transition temperature (*T_g_*) and melting temperature (*T_m_*) of PLA were shifted towards lower temperatures (around 7 °C). Additionally, 90PLA10PCL-15HA and 80PLA20PCL-15HA composites also displayed a double melting peak of PLA, together with a slight decrease in the cold crystallization temperature (*T_cc_*) of PLA.

TGA measurements were performed to further determine the thermal degradation behaviour of the processed materials (Table 3), to investigate the limitation in extrusion and printing temperatures. All samples remained relatively stable without any major weight loss up to around 300 °C, while PCL was stable even up to around 400 °C. Additionally, the composite blends exhibited a residue amount (data not shown) of ~15% compared to the blends, which were completely decomposed.

### 3.3. Mechanical Characterization

Stress–strain curves for all materials are presented in Figure 3 and E-modulus and compressive strength values are reported in Table 4. All blends and composite blends exhibited mechanical properties within the values of native trabecular and cortical bone [36] (Table 4). PLA exhibited the highest ultimate strength, while PCL showed the most ductile behaviour. The curves of both blends and composites remained between those of the pure PLA and PCL. Except for 80PLA20PCL, the addition of HA reduced the ductile behaviour compared to the blend analogues. The compressive strength and the stiffness decreased with increasing PCL content, both for the blends and the composites, except for the 90PLA10PCL blend that differed from the trend.

### 3.4. Accelerated Degradation

#### 3.4.1. Weight Loss

The cumulative weight loss of all samples is depicted in Figure 4. The lowest mass loss was observed for pure PCL (0.6%) after the four-week time period. In contrast, the highest mass loss was observed for the composite blends (~60%), followed by the polymer blends (~30%) and pure PLA (19%). In addition to the weight loss, the degradation caused diameter thinning of all samples (except pure PCL) and the thinning effect increased with increased weight loss.

#### 3.4.2. Chemical and Morphological Evaluation of Blends and Composite Blends

No clear changes in chemical composition could be detected when comparing degraded materials to their pristine analogues. However, a small change in the C=O stretching band could be noted after degradation, as a double peak appeared (Figure 5, black arrows). All blends and composite blends exhibited the same trend, hence, only one of each (70PLA30PCL blend and 70PLA30PCL-15HA) is depicted in Figure 5.

Oppositely, morphological changes at the surface level were clearly visible along with the degradation period (Figure 6). After 6 h (Figure 6, top row), the surfaces of the pure reference materials and the blends were still mainly smooth and with the printing structure, layer-by-layer construct, intact. Some indication of pore formation between the printed layers could, however, be seen for pure PLA and 70PLA30PCL, while some indication of fibre formation on the surface could be distinguished for 80PLA20PCL. In contrast to this, the composite blends exhibited clearly roughened surfaces after 6 h of degradation, which was in accordance with the reported results for the weight loss (Figure 4). After 21 days of degradation (Figure 6, bottom row), more pronounced morphological changes were observed. The printed layers disappeared after 21 days for the composite blends, illustrating the largest morphological changes due to degradation of these samples (Figure 6). More attenuated surface degradation was observed in the blends, while depicting a fibrous and wavier surface compatible with the previous morphology observed for pure PCL. The pure PCL scaffolds’ surface showed the least morphological changes of all materials, illustrating only some pores at the surface level after 21 days of degradation.

XRD data showed differences in the samples before and after degradation (Figure 7). For pure PLA, a major amorphous structure was observed as a neat polymer, whereas after 28 days of degradation, peaks at 19.6° and 22.4° ascribed to α-type PLA [38,39] were seen. Peaks at 21.3° and 23.8° were observed for PCL; degraded PCL samples depicted a broadening in both peaks as a possible reduction in crystallinity (Figure 7A). All the blends depicted the typical α-type PLA, which seemed to increase as the PCL content increased (Figure 7B). In fact, a peak at 16.8° corresponding to this α-crystal in PLA appeared in all of them upon PCL addition and especially intense for 70PLA30PCL. This peak was strongly intense for this blend also when HA was added (Figure 7C). The XRD data from composite blends evidenced the presence of the HA mineral by the peaks corresponding to both HA (Figure 7C, black vertical lines) and monetite (Figure 7C, grey vertical lines). Similar diffractograms were observed for both 90PLA10PCL-15HA and 80PLA20PCL-15HA samples, with the mentioned particularity of the strong α-lactide structure observed for 70PLA30PCL-15HA. After 28 days of degradation, 90PLA10PCL-15HA and 80PLA20PCL-15HA showed a slight increase in the peaks for this PLA structure (peaks at 16.5°, 19.6°), while 70PLA30PCL-15HA after 28 days of degradation depicted peaks only corresponding to PCL.

#### 3.4.3. Thermal Characterization

Degradation effects were observed as a decrease in *T_g_* and *T_m_* for the PLA and the composites, while the PCL and the blends remained similar (Figure 8, Appendix A). The blends and the composite blends also showed a decrease in *T_cc_*, as well as the appearance of two differentiated melting peaks. Furthermore, the composite blends exhibited a higher thermal stability than the blends after the degradation process when looking at the degradation temperature (Table 5). A residue content of ~20% was observed for the composite blends from TGA measurements. No change in decomposition temperature was seen for pure PCL, while PLA showed a significant increase with ~30 °C.

#### 3.4.4. Mechanical Characterization

After 28 days of degradation, both the blends and the composite blends exhibited decreased brittleness when compared to each corresponding non-degraded material, exhibiting a mechanical behaviour similar to that of pure PCL (Table 6). Even though the composites degraded faster than the blends, as seen by SEM (Figure 6) and weight loss (Figure 4), they exhibited higher elastic modulus (except 70PLA30PCL-15HA) compared to the blends after degradation.

## 4. Discussion

In this study, six different filaments for FFF printing were developed and evaluated for printability, physicochemical and mechanical characteristics. Characterizations were performed on non-degraded and degraded materials to evaluate their chemical, morphological, thermal and mechanical properties before and after degradation.

Mixing PLA and PCL resulted in a porous morphology for the blends (Figure 2). Patrício et al. [40] reported that the minor component in a polymer blend most often forms a dispersed phase in the continuous phase formed by the major component. Results from Wachirahuttapong et al. [9], Navarro-Baena et al. [33] and Mattaa et al. [41] are in accordance with the morphologies found in the blends studied. Unlike the blends, the composite blends exhibited a rough surface caused by the mineral phase incorporated in the polymer matrix. Incorporation of HA will, hence, not only increase the bioactivity of the material, the increased surface roughness could also have a beneficial effect on cell adhesion and proliferation [1,17,42,43,44].

Since the thermal properties (*T_g_*, *T_m_* and *T_deg_*) of the blends and the composite blends (Table 3) were detected to lie in between the values of the ingoing pure polymers, blending phenomena of both polymers could be confirmed. The degradation temperatures, for the blends and the composite blends, also verified that the addition of PCL improved the thermal stability of PLA [33]. However, the decrease in *T_cc_* with increasing PCL content might be due to a nucleating effect ascribed to PCL during PLA crystallization, similar to what was reported by Navarro-Baena et al. [33].

Along with the improved thermal stability, an increased onset and offset degradation temperature for both the blends and the composite blends was observed compared to pure PLA (Table 5). On the contrary, Ferri et al. reported that PLA:HA composite blends displayed a slight decrease in onset degradation temperature [13]. Their explanation to the decrease was hydrolysis of PLA, which, most probably, was initiated by the HA since it is a hydrophilic compound with high affinity for moisture. Hydrolytic scission of the chains is the most common degradation pathway for high-molecular-weight polyesters, such as PLA, and can occur through two different main pathways, such as bulk degradation or surface degradation. The former results in a reduction in the polymer molecular weight due to the release of carboxyl and hydroxyl end group by-products. During the latter, surface degradation, the molecular weight stays intact due to the surface by-products leaving the surface by diffusing out to the media, which, instead, gives rise to material thinning [17,22]. However, since this decrease in onset degradation temperature was not seen for the composite blends in this study, it could be hypothesized as an improvement from PCL addition. Overall, the thermal properties indicated that neither the extrusion nor the printing temperatures had a detrimental effect on these materials. This was further supported by the results obtained by FTIR, on the materials as a film, filament or printed, where no changes in chemical composition could be detected. This is of great importance when developing materials for biomedical applications since changes in chemical composition can cause detrimental responses in the body, such as strong immune reactions, for instance [45].

The overall trend seen for the mechanical properties in the blends and composite blends was a decrease in stiffness as PCL content increased, i.e., increasing amounts of PCL increased the ductile behaviour of the materials (Figure 3). For the two blends with a higher amount of PCL, the compressive stress decreased with increasing PCL content, which is in accordance with results from a study by Nishida et al. [18], where they investigated the effect of PCL content in PLA:PCL blends on Young’s modulus and yield stress. Although the general trend was a decreased compressive strength with increasing PCL content, the 90PLA10PCL blend resulted in a lower compressive strength value than 80PLA20PCL and 70PLA30PCL blends. Still, both the processed blends and composite blends exhibited higher mechanical properties than the native trabecular bone but not the cortical bone [36] (Table 5), indicating a possible application as support material for load-bearing cancellous bone applications. Recently, Wu et al. explored the feasibility of using composite filaments for trabecular bone models, which could also be applied to further verify the suitability of PLA-PCL-HA composites as bone substitutes [14]. Moreover, another study investigated the effects of infill density on the compressive strength of 3D-printed pure PLA cylinders. They reported a failure load of 21 kN on cylinders with 80% infill [46], the same compressive failure load as that obtained for the pure PLA with 100% infill density in this study. This implies that a similar resistance to compressive load might be possible to obtain with 20% lower infill density. This can be further investigated on blends and composites to minimize the amount of material to be used.

The weight loss (Figure 4) and morphological analyses (Figure 6) depicted a slower degradation ratio for PCL compared to PLA, correlated to the hydrophobic nature of PCL, and a higher crystallinity [2,10,33]. Interestingly, PLA:PCL blends showed higher degradation rates than pure PLA, in accordance with previous studies [47]. Overall, PLA:PCL blends degraded approximately 10% more than PLA. This phenomenon might be explained by the plasticizer effect of PCL addition, which disrupts PLA crystallization, hence, amorphizing PLA and enhancing its degradation. The changes in surface morphology of the PLA:PCL blends could also have played a role for the higher degradation rates compared to the pure PLA. The blends’ morphology depicted higher roughness with the appearance of pores, which increased in size as the PCL content increased (Figure 2). This morphology, although not specifically measured in this study, could increase the total surface area of the materials and their wettability, thus, promoting a faster degradation. Finally, the addition of HA further enhanced the degradation behaviour, yielding 40% higher degradation compared to PLA. Noteworthily, the inclusion of HA in the polymer matrices might increase the overall hydrophilicity and lower the crystallinity of PLA [2,17], thus, increasing the degradation rates. As for the blends, an increase in the surface area due to the rougher surface in composite blends and the appearance of local porosity, acting as a channel to expose the media to the bulk material, might have also influenced the increase in degradation. Surface degradation was seen by the diameter reduction in the cylinders, accounting for approximately 0.5 mm for the blends and 2 mm for the composite blends (data not shown). Surface degradation was also observed by SEM (Figure 6); the smooth surface from the pristine samples and after 6h degradation became rough at 28d, as an effect of the erosion [17]. It is also possible to see some differences in the surface morphology between the different blends, which, according to Mohseni et al. [2], may be due to different reaction rates of the hydrolysis for the blends.

Despite the significant degradation observed morphologically and with the weight loss in the samples, FTIR spectra did not further evidence degradation by-products (Figure 5). The main by-products from the hydrolysis of PLA are carboxyl (-COOH) and hydroxyl (-OH-) groups [2,17,47], the bands of which overlap with the bands from pristine materials. For instance, the stretching of C=O in the carboxyl group appears around 1700 cm^−1^, while stretching of the hydroxyl group often is detected around 3000 cm^−1^ [48], hence, overlapping with the bands associated to C-H and C=O stretching in PLA and PCL. Only, a small change was detected in the C=O stretching band for the blends and composite blends (Figure 5), depicting a double peak after degradation.

The thermal properties analysed through DSC comparing pristine and degraded samples showed, in general, a reduction in the *T_cc_* and a split in the melting curves for the degraded samples (Figure 8). Lower *T_cc_* has been correlated to a reduction in overall crystallinity and molecular weights [47]. Likewise, the formation of shorter chains during degradation might lead to double-peak formation in melting curves, depicting a first peak for primary polymer structure and a second peak for the new crystal structure [2]. Moreover, composite blends exhibited a decrease in *T_g_* upon degradation, as a potential result from PLA degradation, leading to an increase in lactic acid oligomer by-products, evidenced also by XRD analyses.

PCL was found to increase the thermal stability in the non-degraded materials (Table 5), which could be explained by the faster degradation of PLA, increasing the overall PCL contribution to the thermal properties as PLA hydrolyses. This might also be an explanation for the increased thermal stability in the composites after degradation. Since the PLA degradation was accelerated by the added mineral, the relative PCL content increased compared to the PLA in the original composition. However, the decreased thermal stability seen for the two blends with a higher amount of PLA after degradation could be related to decreased molecular weight [47].

Both the blends and composite blends showed a decrease in E-modulus after degradation, of almost half for all blends and composites, except 90PLA10PCL-15HA (Table 6). This confirms the faster degradation for the stiff PLA, since the materials behaved more similar as pure PCL during the compression tests. Even though the composite blends degraded faster than the pure blends, they still displayed higher elastic modulus (except 70PLA30PCL-15HA) than the blends after degradation (Table 6). This indicates a reinforcement effect of the mineral incorporated into the polymer matrix. However, only the 90PLA10PCL-15HA composite blend exhibited higher compressive strength than the corresponding polymer blend after degradation. Still, the mechanical properties were higher than for the human trabecular bone, indicating that even after some degradation, the material can maintain the mechanical properties needed in order for it to be a candidate for this type of bone replacement.

Overall, composition-customized filaments consisting of combinations of PLA, PCL and HA demonstrated good thermal and chemical stability. The materials showed good printability properties to be applied in a cost-effective FFF-printing technique. The mechanical properties illustrated a reinforcement effect on PLA:PCL blends when HA was incorporated, mostly for 90PLA10PCL-15HA. Further characterization of the degraded materials and released by-products during degradation would be necessary to assess the safety of these materials from a biological perspective and further investigate their potential as bone regenerative materials. Importantly, the degradation behaviour could be actively controlled by the incorporation of both PCL and HA into PLA. Through close control of the amount of PCL, degradation rates could increase up to 10%, while further addition of HA fostered degradation 3-fold for pure PLA or 1.5- to 2-fold for PLA:PCL blends. These degradation rates could be further optimized by the incorporation of porosity during FFF printing and the overall geometry of the scaffold.

## 5. Conclusions

The development of customized materials consisting of biocompatible and biodegradable polymers, alone or in combination with mineral parts, yielded printable materials with chemical stability and mechanical properties suitable for bone regeneration. The mechanical properties were demonstrated to fulfil the mechanical threshold for trabecular bone applications. The customization of degradation rates was highly dependent on the material composition, demonstrating an improvement by the incorporation of mineral phases, such as hydroxyapatite, which further enhanced the degradation of PLA:PCL combinations. This, together with the incorporation of controlled porosity and scaffold architecture, could further offer the possibility of controlling and balancing biomaterial resorption and new bone formation on demand.

## Figures and Tables

**Figure 1 polymers-14-03305-f001:**
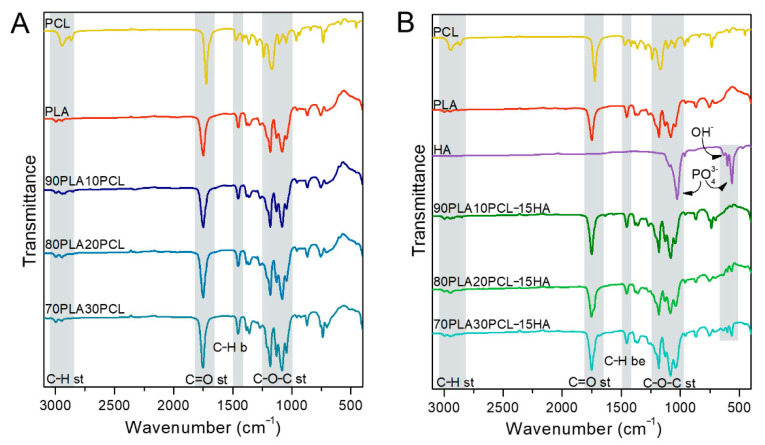
Fourier-Transform Infrared Spectroscopy (FTIR) spectra of pure PCL, pure PLA, pure HA and the processed polymer blends (**A**) and the composite blends (**B**).

**Figure 2 polymers-14-03305-f002:**
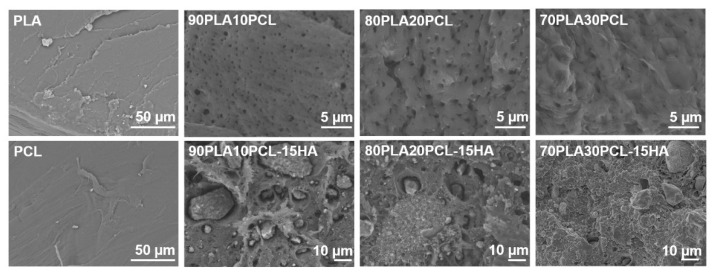
Scanning electron microscopy (SEM) images of filament cross sections for each material.

**Figure 3 polymers-14-03305-f003:**
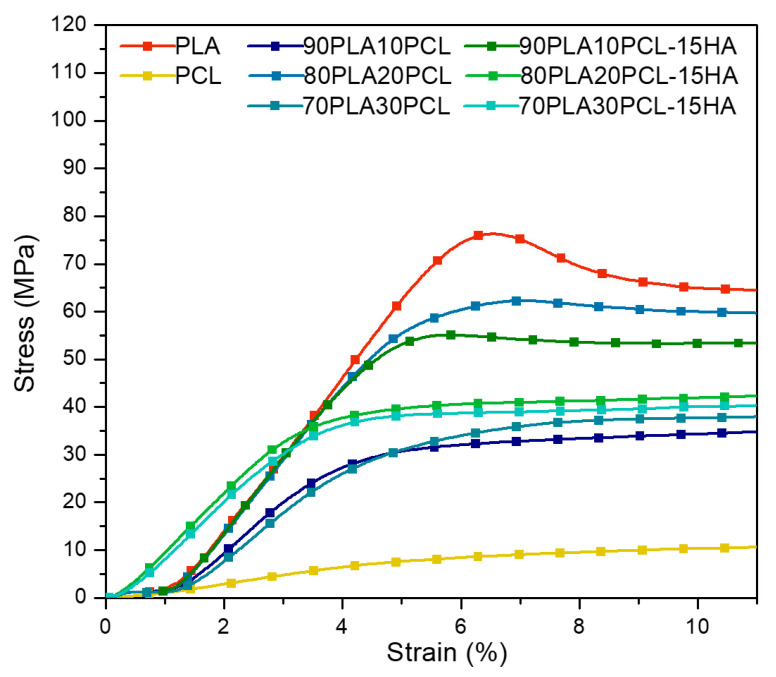
Typical stress–strain curves obtained from compression tests on all materials.

**Figure 4 polymers-14-03305-f004:**
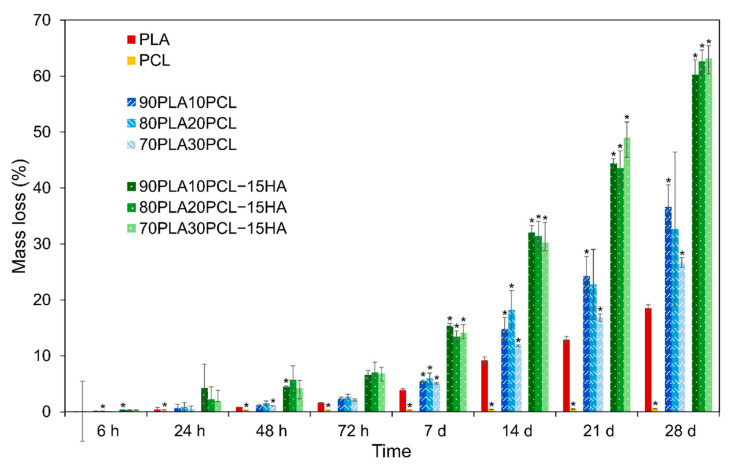
Percentage of mass loss for all materials during the four-week degradation period presented as mean ± standard deviation values. * denotes statistical significance compared to PLA (*p* < 0.05) in the same time point.

**Figure 5 polymers-14-03305-f005:**
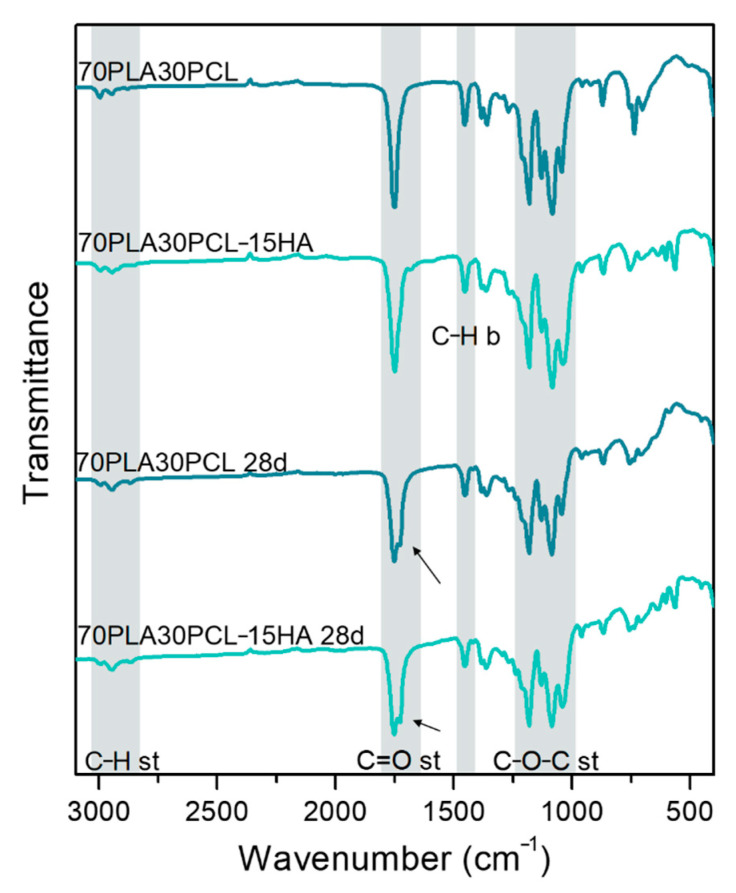
Comparison between IR spectra obtained before and after 28 days of degradation of the 70PLA30PCL blend and the 70PLA30PCL-15HA composite.

**Figure 6 polymers-14-03305-f006:**
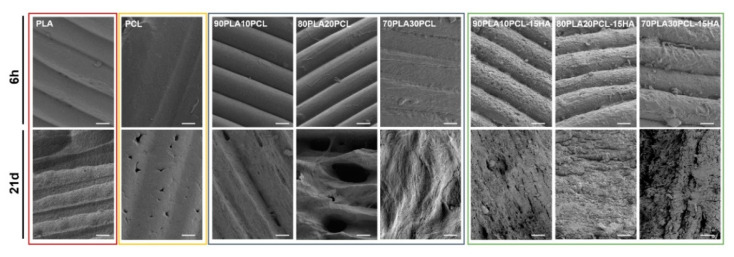
SEM images of the materials after 6 hours (**top** row) and after 21 days (**bottom** row) in degradation media 1M NaOH (scale bar: 50 µm).

**Figure 7 polymers-14-03305-f007:**
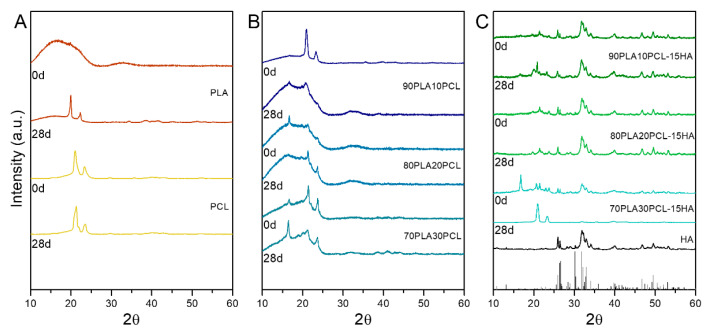
X-ray diffraction (XRD) data for neat samples (0 days) and after degradation (28 days); (**A**) pure polymers PLA and PCL, before and after degradation; (**B**) PLA/PCL blends before and after degradation; and (**C**) composite blends before and after degradation.

**Figure 8 polymers-14-03305-f008:**
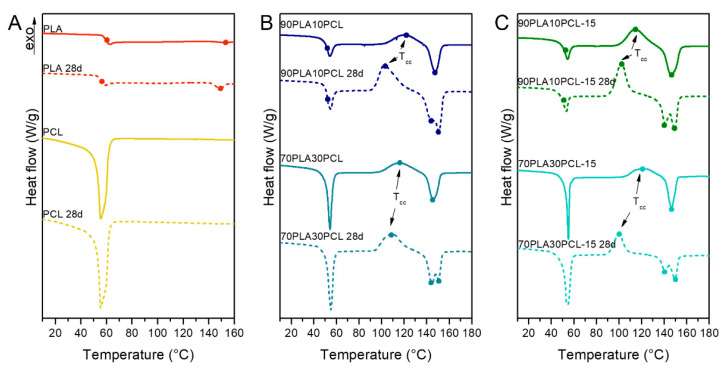
Second heating scan from DSC measurements before and after 28 days of degradation on pure PLA and PCL (**A**), 90PLA10PCL and 70PLA30PCL blends (**B**) and 90PLA10PCL-15HA and 70PLA30PCL-15HA composite blends (**C**).

**Table 1 polymers-14-03305-t001:** Composition of blends and composites describing the amount of polylactic acid (PLA), polycaprolactone (PCL) and hydroxyapatite (HA) in wt%. The designations of the composite samples containing HA were chosen to facilitate the comparison with the samples without HA, as the ratios between PLA and PCL were maintained at 90:10, 80:20 and 70:30.

Designation	PLA (wt%)	PCL (wt%)	HA (wt%)
PLA	100	-	-
PCL	-	100	-
90PLA10PCL	90	10	-
80PLA20PCL	80	20	-
70PLA30PCL	70	30	-
90PLA10PCL-15HA	76.5	8.5	15
80PLA20PCL-15HA	68	17	15
70PLA30PCL-15HA	59.5	25.5	15

**Table 2 polymers-14-03305-t002:** Print settings for the cylinders.

Sample	Print Speed (mm/s)	Nozzle Temperature (°C)	Build Plate Temperature (°C)
Pure PLA	70	200	60
Pure PCL	10	80	30
90PLA10PCL	25	200	60
80PLA20PCL	25	200	60
70PLA30PCL	50	200	60
90PLA10PCL-15HA	50	200	60
80PLA20PCL-15HA	50	200	60
70PLA30PCL-15HA	50	200	60

**Table 3 polymers-14-03305-t003:** Differential scanning calorimetry (DSC) results obtained from the second heating scan and degradation temperature from TGA.

	PCL		PLA				
Sample	*T_m_* (°C)		*T_g_* (°C)	*T_cc_* (°C)	*T_m,1_* (°C)	*T_m,2_* (°C)	*T_deg_* (°C)
PLA PCL 90PLA10PCL 80PLA20PCL 70PLA30PCL 90PLA10PCL-15HA 80PLA20PCL-15HA 70PLA30PCL-15HA	- 54.9 54.2 54.4 54.3 54.4 54.6 54.9		59.9 - 52.5 52.7 53.3 52.9 53.5 54.5	133.7 - 121.2 115.0 116.4 114.9 111.7 122.6	152.6 - 147.6 146.0 145.7 146.4 144.8 146.3	- - - - - 150.5 149.9 -	341.2 410.1 367.5 370.5 368.8 369.4 370.3 366.1

**Table 4 polymers-14-03305-t004:** Elastic modulus and compressive strength for all different materials obtained from compression tests as well as reference values of human bone tissue.

Sample	E-Modulus (GPa)	Compressive Strength (MPa)
PLA	1.66 ± 0.09	77.17 ± 8.37
PCL	0.20 ± 0.07	8.62 ± 2.18
90PLA10PCL	1.09 ± 0.03	31.66 ± 1.05
80PLA20PCL	1.57 ± 0.08	62.29 ± 4.04
70PLA30PCL	1.01 ± 0.07	34.07 ± 3.33
90PLA10PCL-15HA	1.57 ± 0.09	55.07 ± 7.28
80PLA20PCL-15HA	1.26 ± 0.10	39.99 ± 4.66
70PLA30PCL-15HA	1.20 ± 0.07	38.36 ± 0.85

Cortical bone [37]	7–30	100–200
Trabecular bone [37]	0.05–0.5	2–12

**Table 5 polymers-14-03305-t005:** Degradation temperature for the different materials before and after degradation for 28 days.

Sample	*T_deg_* (°C)	
	Non-Degraded	28 d
PLA	341.2	373.9
PCL	410.1	411.9
90PLA10PCL	367.5	352.4
80PLA20PCL	370.5	343.1
70PLA30PCL	368.8	368.5
90PLA10PCL-15HA	369.4	373.2
80PLA20PCL-15HA	370.3	368.5
70PLA30PCL-15HA	366.1	376.7

**Table 6 polymers-14-03305-t006:** E-modulus and compressive strength obtained for all materials before (grey, repeated from Table 4) and after (black) 28 days of degradation.

Sample	E-Modulus (GPa)		Compressive Strength (MPa)	
	Before Degradation	After Degradation	Before Degradation	After Degradation
PLA	1.66 ± 0.09	1.57 ± 0.13	77.17 ± 8.37	83.57 ± 6.42
PCL	0.20 ± 0.07	0.17 ± 0.01	8.62 ± 2.18	11.48 ± 0.99
90PLA10PCL	1.09 ± 0.03	0.59 ± 0.26	31.66 ± 1.05	18.38 ± 6.61
80PLA20PCL	1.57 ± 0.08	0.60 ± 0.08	62.29 ± 4.04	33.80 ± 2.83
70PLA30PCL	1.01 ± 0.07	0.59 ± 0.02	34.07 ± 3.33	31.74 ± 1.61
90PLA10PCL-15HA	1.57 ± 0.09	1.26 ± 0.19	55.07 ± 7.28	34.16 ± 2.94
80PLA20PCL-15HA	1.26 ± 0.10	0.69 ± 0.22	39.99 ± 4.66	26.01 ± 6.81
70PLA30PCL-15HA	1.20 ± 0.07	0.34 ± 0.09	38.36 ± 0.85	4.63 ± 2.52

## Data Availability

The data presented in this study are openly available in DiVA, Uppsala University’s Academic Archive Online, at http://urn.kb.se/resolve?urn=urn:nbn:se:uu:diva-480435.

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
