# Peer review of "The Effect of PCL Addition on 3D-Printable PLA/HA Composite Filaments for the Treatment of Bone Defects"

_polymers, 2022, doi:10.3390/polym14163305_

Round 1

Reviewer 1 Report

The manuscript presents a new composite based on PLA and PCL enriched with HA for bone recovery. The technique used for extrusion of filaments for 3D printing is a fashionable technology because it allows to adjust the filament composition according to the purpose. 

The work is well presented and shows care in the analysis and interconnection in the results. 

I recommend that the work be published with just a few improvements: 

- the line colour of the 70PLA30PCL mixture of both graphs in figure 1.

- calculate the degree of crystallization of the mixtures.

Reviewer 2 Report

The article is well-written and contains interesting results. 

Minor revision is needed: 

My comments: 

- the primer TG and DSC curves (before degradation) should be presented maybe in Supplementary. 

- Fig. 5. the curves are so pale. 

- conclusion is so long, try to shorten. 

Reviewer 3 Report

    The paper provides an overview of the original results of the study of materials used in the process of printing implants for the restoration and healing of bones. The use of polymeric support matrices of polylactic acid (PLA) and polycaprolactone (PCL) has been considered, with the addition of the mineral hydroxyapatite (HA) to enhance biological activity. The material has been shown to have high mechanical properties and a controlled degradation rate, depending on its composition. The results obtained are of undoubted interest for medical applications, since they make it possible to solve the problem of the fragility of currently used materials, as well as to create materials with a controlled degradation rate, which is essential for regenerative tissues. The presented results are quite convincing and well described, and in view of their usefulness, I can recommend this work for publication.

Reviewer 4 Report

In this study, the authors studied the combination of PLA, PCL, and HA composite filaments. The printability for FFF printing, physicochemical characteristics, mechanical properties, and especially degradation behavior of the materials was evaluated. The development of composition filaments consisting of combinations of PLA, PCL, and HA with good thermal and chemical stability suitable for FFF printing is of great significance for enhanced bone repair applications.

The specific comments are as below:

1. English needs to be polished.

2. The design of the "Introduction" still needs to be improved, it is necessary to emphasize the innovative aspects of the research.

3. It is suggested that the conclusion be rewritten.

Reviewer 5 Report

Ref.comments to the paper titled as “The Effect of PCL Addition on 3D-Printable PLA/HA Composite Filaments for the Treatment of Bone Defects” written by the authors: Elin Åkerlund, Anna Diez-Escudero, Ana Grzeszczak and Cecilia Persson

It is known that the search for the materials suitable for use in biomedicine, for example, in bone transplantation and treatment of bone tissues, occupies a special place in materials science, since it is associated with increasing the comfort of human life. Thus, from this point of view, the current paper is modern and actual.

For the first, it is remarked that the authors have made the literature search, analyzing 48 references. Indeed, this will indicates the knowledge of the problem, its useful application and finding ways to solve it. Moreover, the list of the references has included the papers written by the last 5 years, it is very good.

General consideration: The paper is very good!

Little correction: Please use the tilted Latin symbols, for example the value of the weight loss (“… The percentage of weight loss (WL) of all…”).

Experimental details, methods, parts regarded the treatment of the composites and discussion are coincided with our basic knowledge in physics and chemistry.

In the Results section it is very important the data shown in Figure 3. – “Typical stress-strain curves obtained from compression tests on all materials”, in Figure 4. – “Percentage of mass loss for all materials during the four-week degradation period...”, and in Figure 5. “Comparison between IR spectra obtained before and after 28 days of degradation of the…”. It is really supported the possibility to use the novel composite in the bone defects replacement. Nice!

Discussion section is well prepared. Here I would like to ask about the following. The authors showed and confirmed the strength properties of the obtained composites capable of replacing defective parts of bones. This is both Young's modulus and microhardness. And what can be said about the surface abrasion resistance of the material? This is also important when moving a person and when working with hands, feet, and other parts of the body associated with the movement of bones.

Conclusion part accumulates the basic data.

So, the paper is interesting for the specific area for the researchers.

I can recommend to the authors to answer the questions mentioned above.  Thus, the paper can be published after minor corrections.

Reviewer 6 Report

The development of customized materials consisting of biocompatible and biode-gradable polymers, alone or in combination with mineral parts, was found to be a way of further customizing degradation rates and mechanical properties while maintaining the chemical stability of the materials. The topic is interesting; the novelty is average. A careful revision is necessary for reconsideration to publish. The detailed comments on this manuscript are as follows:

My detailed comments are as follows:

1. The language should be polished.

2. Novelty of the work should be established in the section of Introduction.

3. Please make Table 6 more clear.

Round 2

Reviewer 6 Report

After revision, the manuscript can be accepted for publication.